# Dissecting and Interpreting a Three-Dimensional Ground-Penetrating Radar Dataset: An Example from Northern Australia

**DOI:** 10.3390/s19051239

**Published:** 2019-03-12

**Authors:** Lawrence B. Conyers, Mary-Jean Sutton, Emma St. Pierre

**Affiliations:** 1Department of Anthropology, University of Denver, Denver, CO 80210, USA; 2Virtus Heritage, Pottsville 2489, Queensland, Australia; mj.sutton@virtusheritage.com.au (M.-J.S.); e.stpierre@virtusheritage.com.au (E.S.P.)

**Keywords:** ground-penetrating radar, reflection profile analysis, amplitude mapping, burial mounds, northern Australia

## Abstract

A robust 3-D GPR dataset provides interpreters with a variety of methods for extracting important information at buried archaeological sites. An iterative approach that uses reflection profile analysis, amplitude slice-mapping, and often both in conjunction is often necessary as neither method by itself is sufficient. In northern Australia, two constructed mounds contain a number of cultural and geological horizons and features, which can be imaged with GPR. The reflection profiles display the modified ground surface prior to mound construction and some initial construction layers. On the pre-mound surface, amplitude maps of reflective layers that were built-up on the ground surface indicate that they were constructed in an intentional manner. Those surfaces were later covered by sand to produce mounds used for human burial. Human internments in the mound can only be seen in reflection profiles, but once discovered, the profiles can be re-sliced to produce high definition amplitude images of these remains. No one method of analysis can provide an overall interpretation of these complex internal mound features. When the methods are varied, depending on the results of one method, a detailed and varied analysis of certain aspects of the mounds’ internal features are visible, leading to the generation of a number of hypotheses about how this area of northern Australia was used in the past. The 3-D data from GPR shows that this area was an important location on the landscape in the past, and was modified by the construction of a monumental mound, which was then used for human burials, and more recently, the construction of what was likely a ritual enclosure.

## 1. Introduction

The ground-penetrating radar (GPR) method has a unique ability to record reflections of electromagnetic waves in the near-surface within 3-D volumes of ground [1]. When reflection profiles are closely spaced within a grid, hundreds of thousands of individual reflections can be recorded in a small area, leading to a dizzying array of information, which can often overwhelm an interpreter. The method was initially used in archaeology and geological studies by interpreting individual reflection profiles, which are a stacked array of reflection traces that produce images of 2-D vertical slices in the ground [2]. Later, the amplitude slice-mapping technique [2], which is now universally used by most practitioners, was developed to provide important horizontal (or any orientation chosen) images of reflections within designated layers of ground, simplifying interpretation and creating patterns of reflections that can more easily be interpreted [3].

All GPR processing software now commonly allows for large-scale grids of data to be analyzed quickly to produce images of importance in the ground [4] and therefore many recent practitioners have streamlined their interpretive methods and proceed directly from field-collected data to slice-map images after a recommended series of data processing steps are followed. In this common methodological approach, raw reflection data are first given spatial locations, processed to remove background noise, and then further processed in a variety of suggested ways. The reflections are then re-sampled within user provided parameters to create a variety of amplitude images, from which maps are constructed and interpretations made. In this way, extremely complex raw data are quickly made available to the user in understandable images that are often slice-maps or isosurface images [4]. This data processing and analysis approach is extremely popular, especially with many younger geophysicists, as it is much like what I think of as the “smart phone app” approach, where icons can be chosen and the internal workings of software then produce the desired result. The reasoning behind each of the processing steps is often not necessarily understood by the users, and the internal software algorithms that generate the final product are either absent or difficult to access.

This “immediate gratification” method of GPR data visualization is one that most of us now commonly use when starting our interpretation. Here, we suggest that there is important information available within the data that needs to be taken into consideration to provide a more comprehensive and varied analysis of GPR. Elsewhere, the case has been made that there are a variety of data forms that can also provide important information from the ground, which are being overlooked in modern interpretative approaches [5,6,7]. Those include an analysis of individual reflection traces, reflection profile analysis, and data processing specific to individual site parameters. Depending on the questions to be asked, a variety of data analyses should be undertaken while still using the standard amplitude slice-maps as a basis from which to begin.

Here, we provide an example from northern Australia where an analysis of two nearby cultural mounds was interpreted using a variety of interpretation methods in non-standard ways in order to visualize certain internal mound features. Nearby mounds have been studied in a similar fashion, and in those studies, magnetics were also used to determine certain properties of the units visible with GPR [8]. Those results that integrate magnetics and GPR are not repeated here, but instead, details of the GPR methods employed to see what is inside this mound are employed.

In this study, depending on the size of the buried features, their depth and the stratigraphy within the mounds, reflection profile analysis in detailed small areas, large-scale amplitude mapping, and high resolution amplitude analysis with associated profile interpretation were used. The complexity of these mounds necessitated all these approaches, and certain methods were chosen for specific buried features, depending on where they sit in the mound and how they are associated with the internal mound layering.

Shallow remains of a structure were visible using high resolution amplitude mapping within the upper soils. Burials within the mound could be seen in processed reflection profiles, and were only visible when very high resolution amplitude slice-maps were constructed of a very small region of the mound within a specific horizon. More expansive pre-mound units composed of layers that were constructed first on the ground surface, and then later buried under the monument fill sediment, were visible using standard slice-maps. Their extent and geometry were not readily interpretable using the reflection profiles, but their geometry as layered construction units can be seen after frequency filtering and detailed profile construction. The units’ extent and the shape of the fill units were determined by creating amplitude maps directly along the units of importance, which was only possible after they had been identified, corrected for topography, and understood using profile analysis.

## 2. The Site in Northern Australia

For many decades, archaeologists have provided a variety of possible origins for earth mounds in northern Australia, including refuges from floods, seasonal base camp locations, and food preparation and consumptions areas [9,10,11,12,13,14,15,16,17]. There are many hundreds of these prominent constructed features on an otherwise very flat landscape (Figure 1) that vary in shape, size, and presumed functions [8]. In all studies prior to 2017, analyses of the internal components of these prominent features were determined by excavations or coring. The first geophysical analysis of the internal features of these mounds was published in 2018 [8]. The initial goal of that geophysical work was to search for burials within these features prompted by local people who remembered their relatives having been buried there. This was confirmed by the presence of surface artifacts such as spear points that were burial offerings and coral pieces that were commonly used as grave markers, as well as the presence of planted flowering trees such as frangipani and other flowering native flora, all of which are indicative of a burial area [8].

The initial GPR work at many of these mounds identified a number of burials within most, recognized by reflection hyperbolas, found at an appropriate depth for human burials, and which could be identified within at least three parallel GPR profiles spaced 50 cm apart to provide the size and orientation of human remains. Models for what human burials look like in this ground within GPR reflection profiles were obtained by work done at the more formal Mapoon Mission Cemetery (Figure 1) just to the north [18], which occurs in similar ground. Work on a number of the nearby mounds discovered that some were constructed on modified and burned surfaces and only after the mounds had been constructed, or partially built, were they used for human interment [8]. Interestingly, some mounds contain many human burials that are mostly on their northern flanks, while a few appear to be totally barren of whole-body interments.

In general, the Mapoon earthen mounds are between 15–25 m in diameter and average 2–3 m in height, with some reaching 4 m (Figure 2). They are built on and within sand dunes and barrier ridges that are no older than the mid to late Holocene in age, deposited on bauxite-rich sedimentary rocks. The two mounds reported on here include a large one that reaches a 3 m elevation and a smaller one just to the north, both locally called Shadforth’s Mounds (Figure 2).

## 3. GPR Data Collection and Processing

Prior to conducting geophysical surveys, the low vegetation on top and around the mounds was cleared and one grid was established over both mounds, with the topography surveyed using RTK GPS. The GPR data were collected with a Geophysical Survey System model SIR-3000 system (Salem, NH, USA using 400 MHz antennas and a survey wheel for distance calibration. Radar reflections were recorded in a 55 ns time window filtered during collection between 200 and 800 MHz, with 40 traces/m. Reflection profiles were spaced at 50 cm in the grid, and all were corrected for topography.

The reflection profiles were sliced into 6 ns slices (each about 44 cm thick), constructed parallel to the ground surface. This was done to create amplitude images of those features that were built on or near the present ground level. Later in the processing, topography was taken into account, especially when the pre-mound surface was imaged. Each reflection profile was corrected for topography and viewed after the background was removed. No further data processing was initially used in the initial interpretation, but internal features of interest were zoomed in using the profile analysis software [19] to study some profiles in greater detail.

## 4. Analysis of Amplitude Maps for Shallow Features

A first-pass reflection amplitude slice-map created parallel to the ground surface shows the outline of a rectangular structure in the top 44 cm of the ground (Figure 3). When the stones, which are likely pieces of cemented coral, are identified by individual high amplitude reflections, they outline what must have been a small structure or enclosure (Figure 3). It is apparent that this rectangular feature was located between the two mounds. There is no surface evidence for this structure, and it is hypothesized to be a temporary enclosure used for ritual purposes. There are historic photos and plans of enclosures of this sort from elsewhere in northern Australia, and the ethnohistoric literature shows that they were used as windbreaks and sleeping areas and also had ritual functions, such as places for the deceased prior to cremation or burial [20]. There are also reflections visible in the amplitude slice-map from individual stones (coral pieces) within the building (Figure 3), which show no coherent pattern, and their use within this feature is unknown.

To the south of the rectangular feature, also in the top 44 cm of the ground, is a second concentration of coral pieces that generated high amplitude reflections (Figure 3). These objects are located near the crest of the mound, and are likely the result of some other activities there. As the mounds are composed of fine-grained sand, their location at the top of the mound is an indication of human activity, and therefore likely important culturally. Perhaps the top of the mound was paved with coral in the past as a place for people to gather for some kind of important activity. A further analysis of the mound crest feature awaits excavation and analysis.

The interesting rectangular feature between the mounds was identified only using the amplitude slicing method (Figure 3). There are so many large pieces of coral in this shallow slice that their identification and placement in space would have been a laborious process using only a visual analysis of the reflection profiles (Figure 4). Each of these coral objects produces its own reflection hyperbola, and hundreds of them were recorded, with many randomly located. Only after the amplitude map was constructed was the rectangular feature visible, composed of non-randomly aligned stones that made up the wall (Figure 3).

## 5. Reflection Profile Analysis for Burial Discovery

Smaller objects that created reflections deeper within the mounds are very difficult to see in standard reflection amplitude slice-maps (Figure 5). In the slice from an 88–110 cm depth, there are a few high amplitude reflections visible, which are likely produced when the designated slice intersects bedding planes within the mound fill sediment or are areas of concentrated tree roots (Figure 6). None of the reflection features visible within the amplitude map from 88–110 cm is archaeologically interesting.

However, when each of the reflection profiles are viewed and interpreted individually, a number of reflection hyperbolas are visible. These reflections are especially visible after the profiles have been adjusted for surface topography (Figure 6). They cannot immediately be discriminated from the shallower tree root reflections, and were hypothesized to be generated from burials only when placed into space. When these hyperbolic reflections, which are common in the Shadforth’s Mounds are visible in three or more parallel profiles spaced 50 cm apart and in no more than four profiles, they are the length and width of an adult human body. Their depth is also consistent with burials and not the shallower tree roots.

Standard amplitude slice-mapping gives no indication of the burial reflections in the 88–110 cm slice (Figure 5), and they were only visible in detailed profile analysis after being corrected for topography. Once their presence was known, the reflection profiles were re-sampled in a small grid that was 8 × 8 m, only resampling the amplitudes from the depth where the burial was identified in the profiles (Figure 5). Every reflection within the depth slice was sampled and given its own unique location in space, with a 1.2 m search radius used in interpolation during the gridding procedure. The kriging gridding method was also used, which mathematically biases the interpolation values closer to the center of the search radius, producing a more detailed map of the burial feature. The detailed grid shows a distinct northwest-southeast oriented reflection that is about 1.5 m in length, consistent with the size and dimension of a human burial (Figure 5).

For the identification of smaller features within the Shadforth’s Mounds, amplitude slice-mapping alone would have missed these (and many other) burials. The presumed burials were only visible after topographic adjustment of each reflection profile where the hyperbolic reflections were visible at the correct depth for human remains (below the depth of the tree roots). Once their location was approximately known from reflection profile analysis, the reflections within the profiles could be resampled and re-gridded and mapped, showing the correct spatial location of a burial. Fourteen other burials of this sort were visible in these mounds using the same methods.

## 6. Integration of Amplitude Maps and Profile Interpretation

In deeper slices within the mounds, four concentrations of high amplitude reflections are visible (Figure 7) in the 154–198 cm depth, constructed after topography was adjusted so that slices were parallel to the pre-mound surface. These reflection features have no discernable geometry that might indicate a cultural function. All are located at or just above the original ground surface prior to mound construction, with one beneath the smaller northern mound and three under the southern mound. The amplitude map is interesting as it indicates something important at this depth under the mounds, but little else. The reflection profiles show that the high amplitudes were caused by layers of sediment that appear to have been piled up in layers (Figure 8). This is interesting, as the pre-mound features discovered under other nearby mounds with GPR consist of individual objects (coral pieces), which are within a highly magnetic matrix [8]. In some of the other mounds nearby, it was proposed that the concentration of individual objects is mixed with fire ash or other burned materials below the mounds. Those features, which produce remnant magnetic readings, were proposed to be the remains of either cremation rites or perhaps feasting areas where cooking took place [8]. Here, at Shadforth’s Mounds, something else seems to have happened on the original ground surface that involved the layering of some material in four small areas. Perhaps these were constructed surfaces for living quarters, where people built raised areas for everyday activities to get above the wet ground during the rainy season. There are a variety of other ideas about the origin of these surfaces, which must be studied with targeted excavations.

The differences in the pre-mound ground surface prior to mound construction, within mounds just a few kilometers of each other, is interesting [8]. The GPR analysis (accompanied by magnetic mapping of some of the mounds) shows very different types of human behavior that took place prior to the construction of the mounds. In all cases, these areas must have been important for some reason in the past, and they were later transformed into mounds in a monumental fashion and used for the burials of people. This suggests that parts of the ancient landscape perhaps had special locations, where a variety of activities such as cremation, feasting, or living took place. It is possible that the differences are a product of varying activities, or perhaps these areas were used by different people over different periods of time for different reasons. All these ideas need to be refined and tested with the integration of excavation data.

In order to obtain a more detailed view of the pre-mound ground surface and the initial layers of material constructed on it, a standard topographically-adjusted profile was frequency filtered so that only the 400–600 MHz reflections were used to display the smaller reflection features (Figure 8). After this filtering, a new profile was created and the pre-mound surface and about one meter of reflections directly above it are displayed, along the profile where the amplitude map shows the distinct high-amplitude reflections (Figure 8). In this display of reflections the small objects (stones or coral pieces) generate distinct hyperbolic reflections consistent with standard point-source reflections [1]. The pre-mound surface reflection is visible just below these point-source generating objects. The upper surfaces of three distinct piles of material that were placed on the pre-mound surface, covering the objects on it that can be seen, are consistent with piling sediment to create some kind of a living surface or at least elevated area of some sort. Shown in green on the detailed reflection profile (Figure 8), these units are those displayed in the amplitude slice-map from this level within the mound (Figure 7). Later mound construction units can also be seen in the detailed reflections profile crossing this feature (Figure 8).

Only after frequency filtering the raw radar waves and then producing a detailed profile of only the stratigraphy of interest can the origins of the pre-mound units be determined. In this area of the Shadforth’s Mounds, it is apparent that objects of some sort were first placed on the ground surface, and covered by intentionally mounded units to create elevated areas for some reason. Perhaps they were areas for people to keep dry during the rainy season [11] or had some other function that cannot be determined without excavations. Later, this area was transformed into a mound, the remains of which we see today, which was used for the burial of human remains.

## 7. Discussion and Conclusions

At Shadforth’s Mounds, any one method of GPR analysis would have yielded only a partial interpretation of the cultural features below and within the mounds. Amplitude slice-mapping, which has become the standard method for buried feature identification, would have discovered the shallow rectangular enclosure and the general outline of the pre-mound construction layers. The shallow rectangular enclosure was mapped perfectly using this method, with the reflection profiles being so “busy” that using only profiles for analysis would likely have been a failure. The pre-mound layering units would also have been discovered with amplitude slice-mapping, but only profile analyses after elevation adjustments and frequency filtering showed these horizons to be layered units and individual stones.

Smaller features within the mound, such as the numerous human burials (one of which is discussed here), were only visible after individual reflection profile analysis. These burials were effectively invisible using standard amplitude slice-mapping, but a re-sampling and gridding of a small area around one discovered burial showed its exact orientation and size.

These examples from one mound in Australia illustrate that any one data display technique using the 3-D GPR datasets will only produce a limited picture of the ground from which to make interpretations about buried materials. Both amplitude slice-mapping and reflection profile analysis, used in unison and in an iterative way, can provide a more detailed view. This project also indicates how some standard GPR displays must be modified and later reconstructed once something is known about the size and geometry of the features discovered.

Most important for this project is that the GPR analyses indicate that Shadforth’s Mounds was an important place on the landscape of the Cape York Peninsula of northern Australia. When the discovered features are included with those in other mounds studied nearby, it is apparent that people were using this area in an intensive way. Human modification started for a variety of uses, perhaps as areas for feasting, ritual, and everyday activities. Those locations were later transformed into constructed mounds, which necessitated a good deal of coordinated labor, suggesting some authority and motivation by individuals within the societies. After the mounds were constructed, they were used for burials of some, but likely not all, members of society, also indicating social complexity and perhaps incipient social stratification. The mounds continued to be used for burials until recent times, and are remembered by elders as both burial and ritual locations. The shallow rectangular feature discovered at Shadforth’s Mounds is likely the remains of a structure built in the last few centuries, related in some way to a continued ritual use of this important place on the landscape.

## Figures and Tables

**Figure 1 sensors-19-01239-f001:**
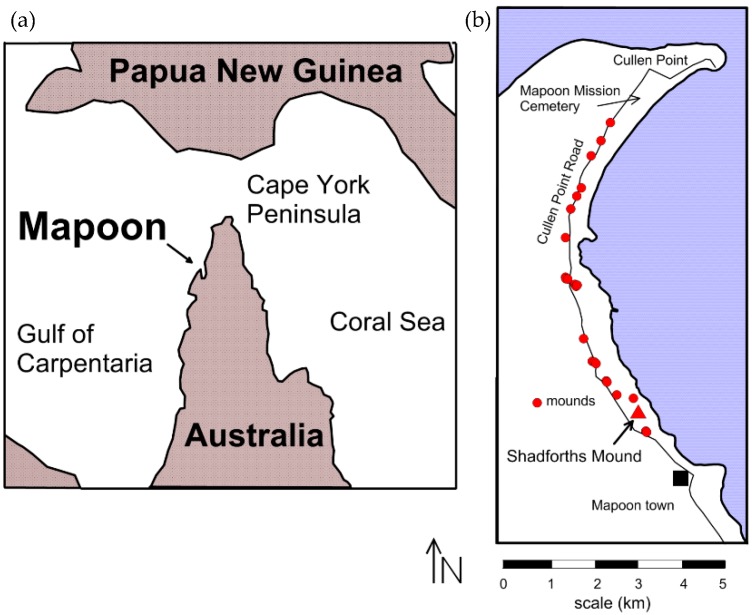
Base map of the Mapoon area, northern Australia, where a number of mounds have been found that are the basis for this study. (**a**) Mapoon location in northeast Australia (**b**) Mound locations near Mapoon.

**Figure 2 sensors-19-01239-f002:**
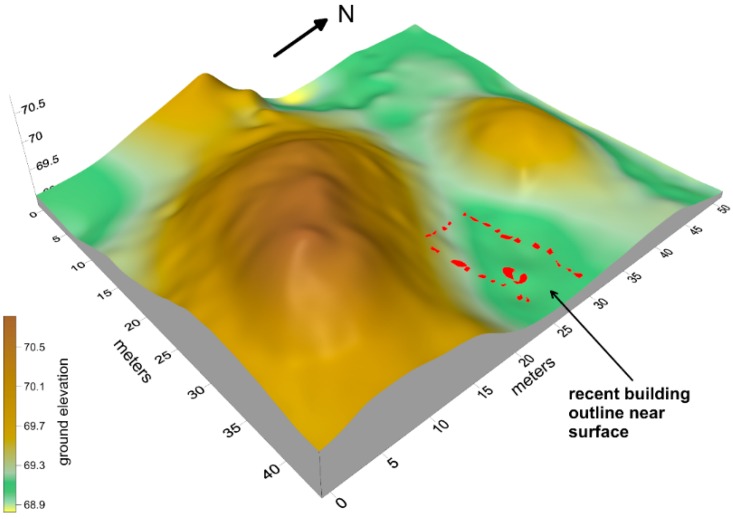
Topographic image of the Shadforth’s Mounds showing the location of the recent structure between the north and south mounds.

**Figure 3 sensors-19-01239-f003:**
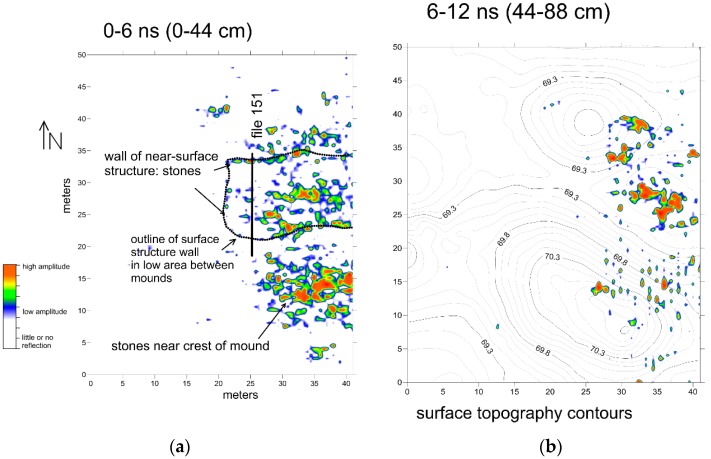
(**a**) Amplitude slice-maps of the first two slices of the ground, showing the outline of the near-surface structure and other stones near the southern mound crest. (**b**) The surface topography contour map is overlaid on the amplitude map on the right.

**Figure 4 sensors-19-01239-f004:**
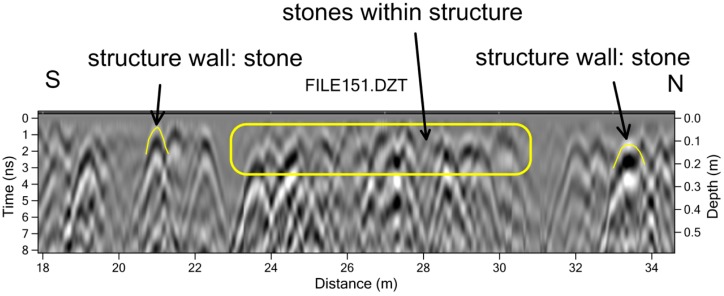
Reflection profile showing the upper 50 cm of ground with some of the near-surface stones that define the walls of the structure noted, but many other stones that are randomly located.

**Figure 5 sensors-19-01239-f005:**
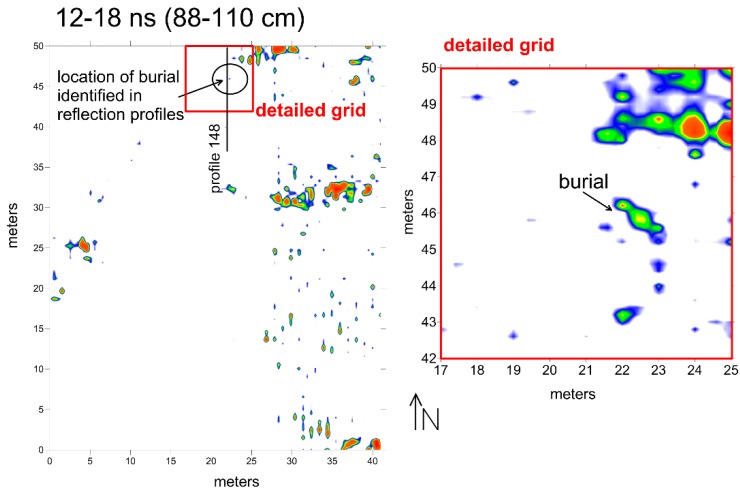
Amplitude slice-map of the 88–110 cm level showing where the burial reflections should be in standard slice-map production, and how re-sampling and gridding of a small detailed area of the grid can show the human-shaped burial reflection in greater detail. Location of file 148 in Figure 6 is shown.

**Figure 6 sensors-19-01239-f006:**
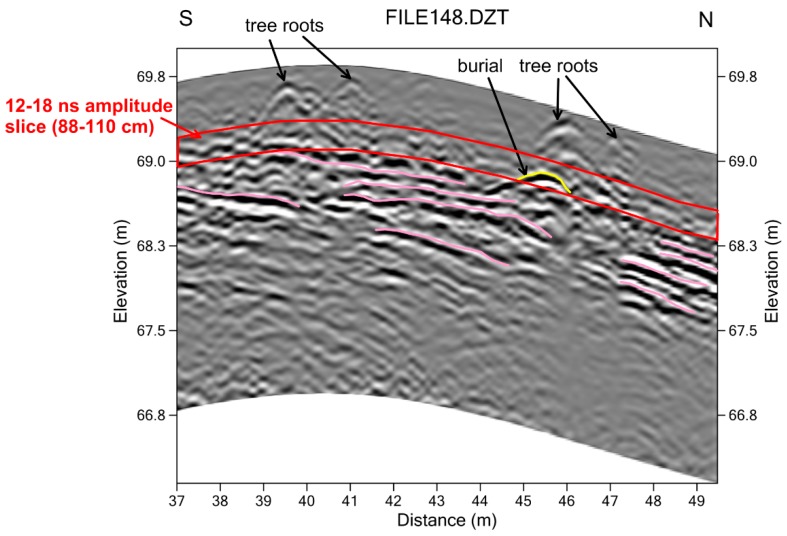
Reflection profile 148 (location shown in Figure 5), corrected for topography, showing the reflection hyperbola produced by a human burial below the tree-root reflection. Mound fill units are shown in pink.

**Figure 7 sensors-19-01239-f007:**
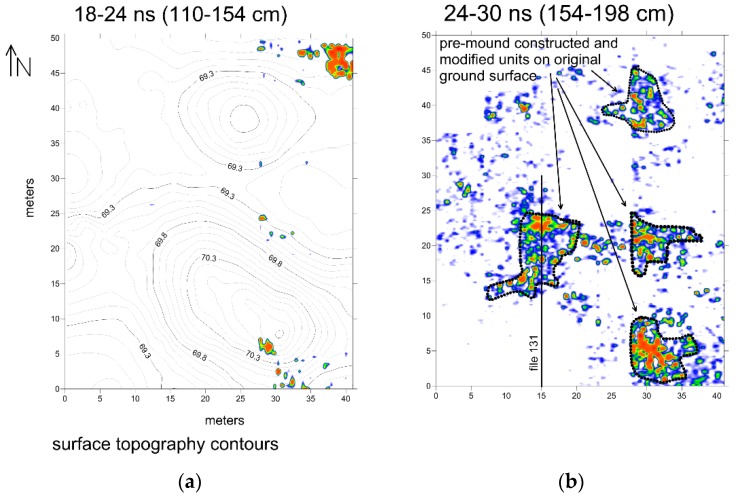
Amplitude slice maps of the two deepest slices at Shadforth’s Mounds showing the concentrations of high amplitude reflections on the pre-mound surface in the deepest slice from a 154–198 cm depth. The surface topographic contours are superimposed on the 110–154 cm depth slice. (**a**) Topographic Contour map; (**b**) High amplitude concentrations on the pre-mound ground surface.

**Figure 8 sensors-19-01239-f008:**
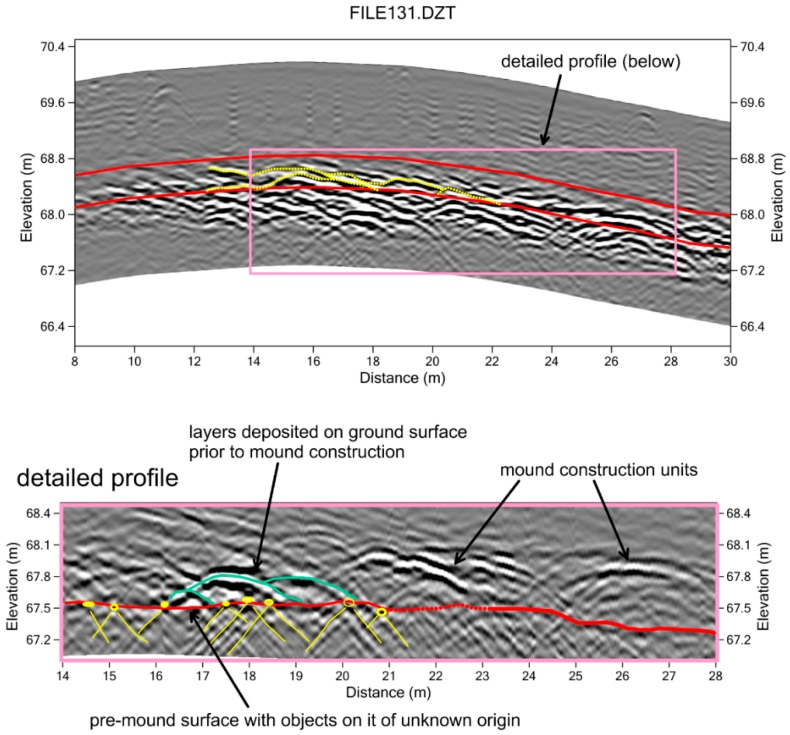
Reflection profile that images the materials placed on the pre-mound surface, with a detailed profile of one part of the surface, after the reflections were frequency filtered to include only those between 400 and 600 MHz, which removed some of the broader planar reflections and enhanced the point-source reflection hyperbolas (axes of which are shown in yellow).

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
