# Peer review of "Dissecting and Interpreting a Three-Dimensional Ground-Penetrating Radar Dataset: An Example from Northern Australia"

_sensors, 2019, doi:10.3390/s19051239_

Reviewer 1 Report

I think the premise behind the paper is good, as it involves Aboriginal community participation, ethnography, archaeology and GPR to understand burials in earthen mounds more accurately. The authors state this approach is holistic, but I think that needs to be highlighted more throughout the paper. This paper does not come across very holistic. Other methods that assist in the GPR interpretation are mentioned but the evidence to support their conclusions is lacking. For instance, an image of what a structure might look like in the past could be helpful in seeing how this might be represented in the GPR data, or incorporating the magnetic gradiometer results to that of the GPR to show how the two instruments are similar/dissimilar. Additionally, the concept of resampling and gridding the GPR data has been completed elsewhere in Australia on Aboriginal burials so it would be worthwhile that the author's acknowledge the success of that study, which is also holistic as well. 

Author Response

Reviewer 1 was under the impression that this paper was about graves of Aborigines, which it is not.  I and my co-authors have published 3 papers already on that theme in three different journals. So it would be wrong to go over that same ground again, as it would be just repeating what was done already.  Instead this paper is about how to dissect a GPR dataset to get different information and interpretations using different techniques.  And I have made the case here that those techniques are often ignored by modern GPR people who think that their powerful software can answer all their questions with no real work or thought on their part.  That is what this article is about, and I just happened to be using a really good dataset from an Aborigine burial mound to make these points.  So I removed all the words “holistic” from the paper, which seemed to set off Reviewer 1 to account for what I really did.

1. Page 2: I think you need to credit Lowe et al.'s 2014 work on mapping Aboriginal burials in a rockshelter in Australia. Their approach was also very holistic and it involved resampling and re-gridding their GPR data to determine the location of burials, something that was difficult to detect in the original GPR data.

LC:  I will cite this below in another article we are writing about this site.  This article is not about burials, but about the methods of data processing and analysis.

Lowe, K.M, et al. 2014 Ground-penetrating radar and burial practices in Western Arnhem Land, Australia. Archaeology in Oceania 49: 148-157.
2. Page 4: How do you know this is a structure without testing it? Was there a floor or compacted surface inside the structure? Can an image of one of these structures be provided?

LC:  I have an image of what historic structures like this are like, but have no permission to publish them because of cultural sensitivity and no copyright to obtain.  So I have cited the book where these are illustrated instead.  And we are NOT testing any of this right now, also because of cultural sensitivity.  

3. Page 7: Figure 6, Is this a burial too?

LC: could be, but this article is not about burials, so I am not  trying to report on all the burials that might be there.

4. Page 7: Could it be bedrock?

LC:  Yes...but I have a tone of profiles of what the bauxite bedrock looks like in this area, and these are layered units on top of that bedrock, not bedrock per set.  

5. Page 7: How deep were these anomalies detected? Mag has a maximum depth generally the distance of the sensor spacing which would be 1 m.
If the mounds are taller than that, how do you know if pre-mound surfaces were burned?

LC:  here I am talking about other mounds that are not as tall as this.  That is all published elsewhere already.  But in case you want to know this is not a magnetically high area.  It is bauxite that has no remnant magnetism, with all iron and other magnetic materials leached out of the system.  Overlain by coral and quartz sand.  So any magnetic values that are collected are showing something very different than the "background".  

6. Page 7: Were these tested to verify this? Unclear.

LC:  no.  No testing allowed here.

7. Page 8: So if targeted excavations are being proposed, why not wait until that happens to confirm your observations?

LC: that could be years in the permission process.  How I would love to get out there and dig all this up.  But this article is not about digging results but about the GPR processing methods.  We will have to wait some time to get these direct correlations.

8. Page 8: Figure 7, Perhaps add an outline of the mounds?
9. Page 8: Figure 8, This reflection profile looks different than the one on top.

LC:  that is because the profile has been filtered to remove many of the horizontal planar reflections and enhance the point-source hyperbolas.  I have made this notation in the figure caption, as another reviewer also commented on this.

10. Page 9: It would be good to see the magnetometer results since it is mentioned quite a bit in this paper. It would enhance the current results and verify some of the observations made by the authors.

LC: ditto that above.  We have already published that and this paper is not about magnetics.  Only GPR.

11. Page 9: Again Lowe et al. 2014 did a similar processing analysis (re-sampled and gridding) on Aboriginal burials in a rockshelter.

LC:  ditto also that above.  I will cite this in one of our next papers on this area.

12. Page 10: I am not sure this is holistic, I think processing both ways is pretty standard in GPR.

LC:  I have taken out the word holistic.  Was perhaps getting ahead of myself.  And how I wish that this kind of data processing and analysis was "pretty standard".  I have been ranting and talking now about how this needs to be done more for 10 years, and as the editor of a journal I was consistently let down by authors who did NOT do this kind of analyses.  So I hope that this reviewer and I can team up and try to get the GPR community to pay attention to all these details in their data and not just run through some processing steps in the hope that results will just appear.  So how about we go for it in this endeavor?

13. Page 10: Community would be a better word choice here."

LC:  yes...that is a good word.  Thanks.

Here is that other paper that has been published already on the burials and the magnetics:

2018: Integration of GPR and magnetics to study the interior features and history of earthen mounds, Mapoon, Queensland, Australia.  With Emma J. St Pierre, Mary-Jean Sutton and Chet Walker.  Archaeological Prospection, v. pp 1-10.  DOI 10.1002/arp.1710

Reviewer 2 Report

The most comments are placed in the pdf manuscript.

Author Response

Notes on changes made based on Reviewer #2’s suggestions:

1.      Optionally: 3D-GPR data: I kept the title with three-dimensional, as I don’t think it is appropriate to use abbreviations in a title.  But then changed to 3-D (also 2-D) throughout the rest of the manuscript, as suggested.

2.      "Database" is not a correct word in this context. Perhaps is better "dataset": Database has been changed to dataset.  Good suggestion!

3.      In my opinion, this map cant be rotatein order to reduced the figure area: Figure 1 has been changed so as to not use up so much space.  New figure is attached called 1.png

4.      This units are IS, so it is not necessary explain it.: All nanoseconds have been changed to ns as suggested, which is good.

5.      Other abbreviations also have been made to change centimeters to cm etc.

6.      Why the authors not explain  the A and B reflections sets? : I have not changed Figure 3 to try to explain every single anomaly in it, as that was not the point.  Only to show how the one non-random structure could be seen.  Doing more than that will just clutter the manuscript, and I am trying to only point out some features, not all.

7.      why not these others?... It is not  concluyent In my opinion the 151 file is not representative, the authors could by presented jointed the suggested profile market in figure 3a, croissing the A abd B datasets; in order to better compare. : I like Figure 4 (file 151) as it shows how complicated all the reflections are in the ground, and how only some of them are important.  So I have changed the manuscript to point this out, and try to explain why that figure is so important.  The conclusion is that trying to pick this feature out by looking at profiles will only be a failure, as there are so many reflections!

8.      It could be to place the profile in the slice map in figure 5 : I put the location of file 148 in Figure 5 as suggested, and put a note in the caption that it is there.  Then changed the caption for Figure 6 (which is file 148) to note that its location can be seen in Figure 5.  Thanks for catching that!  A new figure 5 is attached as 5.png

9.      B: In Figure 5 I just can’t deal with trying to explain every anomaly in the slice-maps.  So I have not changed it to explain that other feature, but in the text said that it is a product of the slice crossing a stratigraphic boundary.

10.  Why not this signals too??: Figure 6:  YES…there is lots of other stuff going on here, and some of those other reflections may or may not be important.  But trying to explain it all is just too much information for this short paper.

11.  The authors could indicate, briefly, as the feature affects in the GPR signal. : I don’t want to get into the magnetics correlated to the GPR, as we just published a paper on that subject, which is cited here.  But I did make more explicit what types of magnetic features these are (remnant magnetism), in the text.  That was a good suggestion!  The readers will have to go to that other paper of ours to read about the magnetics and GPR correlations.

12.  It could be interesting to put the topography map in this slice: Figure 7:  I put the topography on the map on the left, so that the really complex slice on the right is not too “busy”.  The readers can look to the left and see where these site in the topography of the site.

13.  It seams that is not the same detailled profile; Explain why you market this signals; The central frequency of antenna is 400 MHz, so this band is at the higth range.... why the authors select this band? In other hand, the radragram not looks that it was to be filtred whith at 400-600 MHz band-pass filtrer: Figure 8:  the lower profile really is only 400-600 MHz, filtering which removed some of the long planar reflections and enhanced the reflection hyperbolas, so that I can see each individual stone on the pre-mound surface.  That is the miracle of frequency filtering, if you do it cautiously.  In the caption of Figure 8 I made sure to explain what the yellow axes of these reflection hyperbolas are (noting the individual locations of stones on that surface)

14.  It my be interesting to present a figure whith the sintesys of this magnetic-gpr study; In my opinion the radargrams presented not seams to be filtered whith 400-600 MHz band-pass nfilter: The note that I should talk more about other mounds nearby in this statement: accompanied by magnetic mapping of some of the mounds.  I made sure that the reader knows to look at the other recently published paper to see this, and placed that citation again here.

Thanks Reviewer 2 for the great suggestions and edits.  This has made this a much better paper. 

Reviewer 3 Report

The free area on the left of Figure 1 must somehow be corrected.
It would be interesting to show diachronic images, e.g. from Google Earth, in the case which you will find marks of covered constructions. Furthermore, this would be more helpful in understanding the study area.

Author Response

Reviewer 3 only suggested that I put in some image from Google Earth of the Mapoon area.  I spent some time on that and found that the satellite coverage of this area of northern Australia is poor, and none of the images were worthy of using. So I did not change Figure 1 from what I have originally submitted.

Round  2

Reviewer 2 Report

In this second revision the basics coments are: 

= Some of the formal comments are not revised

= The processing flow is poor. The authors say that the general 3D interpretation was  make with raw GPR data without topographic corrections, and only for a particular interpretive aspects they processed some GPR profiles and apply this correction. If we  consider the significative toopography of the study zone, it is necessari processed all 3D data as a similar mode as the presented processed  profiles. Only then it does  sense generate  the slices images and 3D general views (volumtric images). 

Author Response

= Some of the formal comments are not revised

I didn't take some of the comments and work them into the last version, as I didn't think them appropriate for what I was trying to show, which is how multiple methods for processing must be used.  I did, however, comment to the reviewer my take on those comments. 

= The processing flow is poor. The authors say that the general 3D interpretation was  make with raw GPR data without topographic corrections, and only for a particular interpretive aspects they processed some GPR profiles and apply this correction. If we  consider the significative toopography of the study zone, it is necessari processed all 3D data as a similar mode as the presented processed  profiles. Only then it does  sense generate  the slices images and 3D general views (volumtric images). 

  This is not an article about processing flow, and in fact, I am a critique here in the usual "flow of processing" that some software users demand.  I find that any processing flow is following the "smart phone app" method of data processing, which I have commented on and reject.

On topographic corrections.  The initial slices that were created were NOT topographically corrected, as the volumes needed to follow the present ground surface.  That was important especially with that shallow structure, which was built on or close to the present ground surface.  But of course topography is important here, and therefore it was taken into account for the amplitude maps of the ground surface prior to slicing and producing those images of the features on that surface before mounds were constructed.  If that was not clear, then I will go back right now and look at that wording, making sure it is in there.  And of course, topographic variations were taken into account for the profile analyses, as can be seen in the figures.  That is very important!  So the reviewer is very correct in that regard.  But this again shows how a "flow" of processing that would usually be done in "standard" methods is not appropriate here.  So that is the theme of this paper.  I am attaching a revised manuscript that make sure the readers will note that topography was taken into account prior to the construction of the slices of the pre-mound ground surface.

NOTE:  here is what I said on the processing flow:  The reflection profiles were sliced into 6 ns slices (each about 44 cm thick), constructed parallel to the ground surface.

And so I added this to make sure it was clear what I did on this "non-standard processing flow:This was done to create amplitude images of those features that were built on or near the present ground level.  Later in the processing topography was taken into account, especially when the pre-mound surface was imaged. 

And then I added this also, to make sure this more standard topographic adjustment processing step is clear:

In deeper slices within the mounds four concentrations of high amplitude reflections are visible (Figure 7) in the 154-198 cm depth, constructed after topography was adjusted so that slices were parallel to the pre-mound surface.

New manuscript is uploaded with these revisions.